# Spatial distributions and determinants of intimate partner violence among married women in Ethiopia across administrative zones

**Meseret Tadesse Fetene[1], Senait Cherie Adgeh[2], Haile Mekonnen Fenta[2]***

1 Department of Psychiatry, College of Medicine and Health Science, Bahir Dar University, Bahir Dar, Ethiopia, 2 Departments of Statistics, College of Science, Bahir Dar University, Bahir Dar, Ethiopia

* hailemekonnen@gmail.com

## Abstract

### Background

Intimate partner violence (IPV) against women is highly prevalent in the world, especially in low-middle-income countries including Ethiopia. Studies so far assessing risk factors for IPV often use the classical model without geographical location information and spatial effects. This study aimed to estimate the overall prevalence and associated risk factors of intimate partner violence among Ethiopian administrative zones.

### Method

The 2016 Ethiopian Demographic and Health Survey (EDHS) data were used. The primary outcome of the study was the experience of different types of IPV: physical, emotional, and sexual by ever-married women aged 15–49 years. We adopted a generalized multilevel mixed model with IPV as an outcome variable and zones as random effects.

### Results

The prevalence of physical, emotional, and sexual violence in Ethiopia are respectively 20.38%, 22.31%, and 7.58%. The result indicated that 1,423 (30.15%) of respondents had experienced at least one of the three types of IPV. Women who had older age had more children, had lower decision-making power, and had a husband who was a drinker and had controlling behavior were more likely to experience any forms of IPV. Significant zone-wise spatial variations of IPV were also observed.

### Conclusions

The distribution of IPV in married women varies among Ethiopian administrative zones. Several factors were associated with IPV, therefore, interventions targeting the hotspot areas and specific determinant factors should be implemented by the concerned bodies to reduce IPV among married women in the population.

**Data Availability Statement:** The data we used is the DHS, which is the '2016 Ethiopian Demographic and Health Survey' were obtained

from the DHS program http://www.dhsprogram.com but the 'Dataset Terms of Use' do not permit us to distribute this data as per data access instructions. Any researcher can access data after becoming an Authorized user. To get access for the dataset researchers must first be a registered user of the website and access permission has been provided, users may download the 2016 Ethiopian Demographic and Health Survey. In addition, the shape file of the map was freely available from https://africaopendata.org/dataset/ethiopia-shapefiles." Please see the revised manuscript.

**Funding:** The authors received no specific funding for this work.

**Competing interests:** The authors have declared that no competing interests exist.

**Abbreviations:** EDHS, Ethiopian Demographic and Health Survey; SNNP, South Nation Nationalities and People; AOR, Adjusted Odds Ratio.

# Background

Intimate partner violence (IPV) is defined by the World Health Organization (WHO) as "any behavior within an intimate relationship that causes physical, emotional or sexual harm" [1]. Different literatures use the terms IPV: as gender-based violence, violence against women, women domestic abuse, domestic violence, and spousal violence interchangeably [2–4]. It is a fundamental violation of women's human rights, as well as a significant public health crisis, and a major obstacle to sustainable millennium developmental goals [5]. Even though IPV exists in all societies, its prevalence varies widely by areas [6, 7]. Globally, almost one-third (30 to 38%) of all women who have been in a relationship have experienced physical, emotional or sexual violence and 38% of all murders of women are committed by intimate partners [6, 8]. But the report from the 2010 Global Burden of Disease, the largest prevalence of IPV is found in central sub-Saharan Africa, with 65.6% of women have experienced it [6, 9]. A meta-study conducted from international demographic health surveys (DHS) between 2010 and 2017 across 46 low and middle-income countries, including Ethiopia, among women aged 15–49 years reported varied prevalence across countries from 3.5% in Armenia and Comoros to 46% in Afghanistan [10, 11]. But studies in Ethiopia systematically reviewed, the lifetime prevalence of physical, sexual, and emotional violence ranged from 31 to 76.5%,19.2 to59%, and 51.7%, respectively [12].

Research in developed countries showed that women exposed to IPV are more likely to experience symptoms of depression, anxiety, psychogenic non-epileptic seizures, and psychotic disorders [13]. In a study done in South Africa among women, who had IPV, 45.3% reported clinically main symptoms of depression and 30.0% suicidal ideation [14]. Association between IPV and mental illness have been reported in different literatures [15–17]. Women experiencing physical violence, childhood sexual abuse, emotional violence, and spousal control were factors independently associated with depressive episodes [17]. Common risk factors for IPV include: low levels of education, exposure to child maltreatment, witnessing parental violence, antisocial personality disorder, substance abuse, multiple partners/infidelity, attitudes accepting of violence [8], gender inequality, and poverty [18], early marriage [19, 20], younger and not empowered women, and those living in rural areas are more vulnerable to IPV exposure in most countries [11]. The risk factors identified from a limited number of studies in Ethiopia include; older women, were married before the age of 18 years, witnessed inter-parental violence during their childhood, had a partner who drank alcohol, and living in a community with high IPV-accepting norms [21], living in rural areas, divorced, primary and secondary education, 25–39 years old, being poor are vulnerable to IPV [22]. Spouses have a positive attitude towards women's autonomy, educated men and men who have higher access to information are less likely to perpetrate violence [23].

Due to its profound and long-lasting consequences for survivors and families, the international community has increasingly recognized the immediate attention to improve global policy action to avoid violence against women [24]. Despite the growing international attention, however, there is still limited investment in IPV research and coordination in measuring progress toward the 2030 SDGs in most LMICs [25]. More investments in research to build evidence based on the associations between IPV and its factors are essential for policymakers. Though several studies demonstrated that Ethiopia has recorded promising progress in reducing levels of IPV over the past decades, the challenges and achievements of different administrative Zones have not been studied yet. Hence detecting the problem of IPV and its variation among administrative Zones provides deeper insight into the country's health priorities for IPV among women for zonal health departments to plan, follow up, monitor, and evaluate health activities at the lower level which helps policymakers to design focused intervention

strategies. There are some cultural variations among administrative Zones, which result in different practices regarding IPV at the Zone level [26–32]. We therefore set out to determine the prevalence and risk factors for intimate partner violence and look for its distribution at different regions and administrative zones of Ethiopia using the most recent EDHS 2016 dataset.

## Methodology

### Study design

The data for this paper was drawn from the nationally representative cross-sectional study design, 2016 Ethiopian Demographic and Health Surveys (EDHS), described in detail at https://dhsprogram.com and we used the shapefiles of the second administrative zones of the country (https://africaopendata.org/dataset/ethiopia-shapefiles). In DHS, multistage sampling was used to select the sample for each survey: where the first step of the sampling procedure involved the selection of clusters (enumeration areas (EAs)), followed by systematic household sampling within the selected EAs. The number of clusters is the first stage which is selected from the list of enumeration areas (EAs) created in the recent population census of the country and the households that are randomly selected in each of EAs. From the selected households, women aged 15–49 years are selected for an in-depth interview [33]. In the first stage, 645 primary sampling units were chosen in the first stage (443 from rural areas and 202 from urban areas). In the second stage, an average of 28 households from each primary sampling unit were selected through systematic random sampling. The data focused on individual records from ever-married women aged 15 to 49, with 15,683 participants achieving a 95% response rate. For the domestic violence module, one married woman per household was interviewed, resulting in 5,860 women being selected and interviewed with a 97% response rate. The current study analyzed responses from a weighted sample of 4,720 ever-married women who completed the intimate partner violence questionnaire. Sampling weights were adjusted to account for the complex sampling procedures and ensure the results are nationally representative [34].

### Inclusion and exclusion criteria

This study was focused on the experience of intimate partner violence (IPV) among ever-married women in Ethiopia. It included all ever-married women aged 15–49 in the last five years preceding the surveys in the enumeration area who are currently married, divorced, or widowed. Exclusion criteria include missing values and respondents younger than 15 or older than 49.

Ethiopia is situated in the Horn of Africa from 30 to 140 and 330 to 480 E. The country is a low-income country in East Africa, is a completely landlocked country that has a surface area of 1.1 million km$^2$ [35]. For administrative purposes, the country is divided into 11 regions; and a total of 72 administrative areas called zones, a setting for which the entire analysis is carried out (Fig 1).

### Variables

The outcome variable for this study includes single or multiple forms of physical, emotional, and sexual partner violence which was assessed using women's self-reported responses to the questions depending on the modified Conflict of Tactic Scales of Status [36]. This information is crucial for understanding which women have experienced multiple types of violence in their lifetime. The emotional, physical, sexual, and any combinations of violence had a Cronbach's alpha of 0.82, 0.67, 0.72; and, 0.68 respectively, indicating overall good test performance of the interview questions (Table 1). The IPV is defined as women who have experienced at least one event of physical, emotional, or sexual violence since the age of 15 years [25, 36, 37]. The

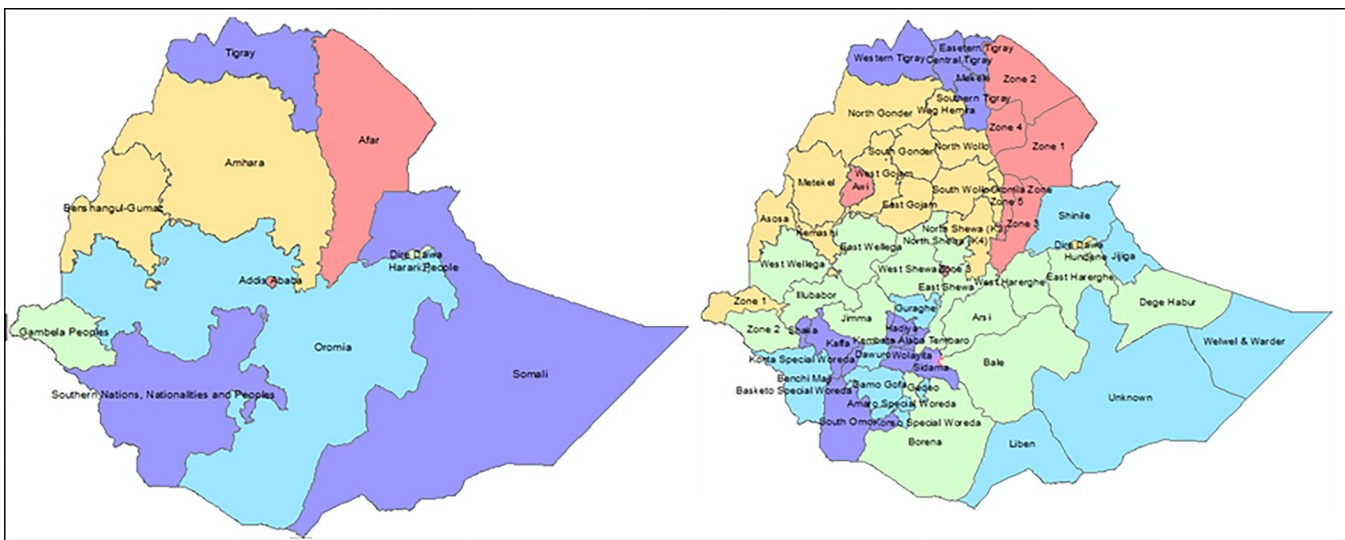

**Fig 1. Map of Ethiopia and showing regions and zones.** A) The eleven regions of Ethiopia; B) The 72 zones of Ethiopia.

covariates include all socio-demographic characteristics that were taken as the independent variables for the different survey years, which were selected from other kinds of literature [5–8, 10, 11, 16] (Fig 2).

## Spatial heterogeneity of IPV prevalence

First we computed and mapped the crude prevalence of IPV in the 72 administrative zones. The prevalence of IPV was estimated at each zone and we used ESRI Desktop 10.3 [38] to generate the maps of IPV prevalence using a kriging interpolation technique, a methodology widely used in spatial mapping [39–41].

## Statistical analysis

The data management was done using Stata version 16. The data were weighted to make them representative and to provide better statistical estimates. We adopted the generalized linear

**Table 1. The tool used to measure IPV in the demographic and health surveys.**

| Question/item | IPV type |
|---|---|
| Push you, shake you, or throw something at you? | physical IPV |
| Slap you? | |
| Twist your arm or pull your hair | |
| Punch you with his/her first or with something that could hurt you? | |
| Kick you, drag you, or beat you up? | |
| Try to choke you or burn you on purpose? | |
| Threaten or attack you with a knife, gun or any other weapon? | |
| Physically force you to have sexual intercourse with him even when you did not want to? | Sexual IPV |
| Physically force you to perform any other sexual acts you did not want to? | |
| Force you with threats or in any other way to perform sexual acts you did not want to? | |
| Say or do something to humiliate you in front of others? | Emotional IPV |
| Threaten to hurt or harm you or someone close to you? | |
| Insult you or make you feel bad about yourself? | |

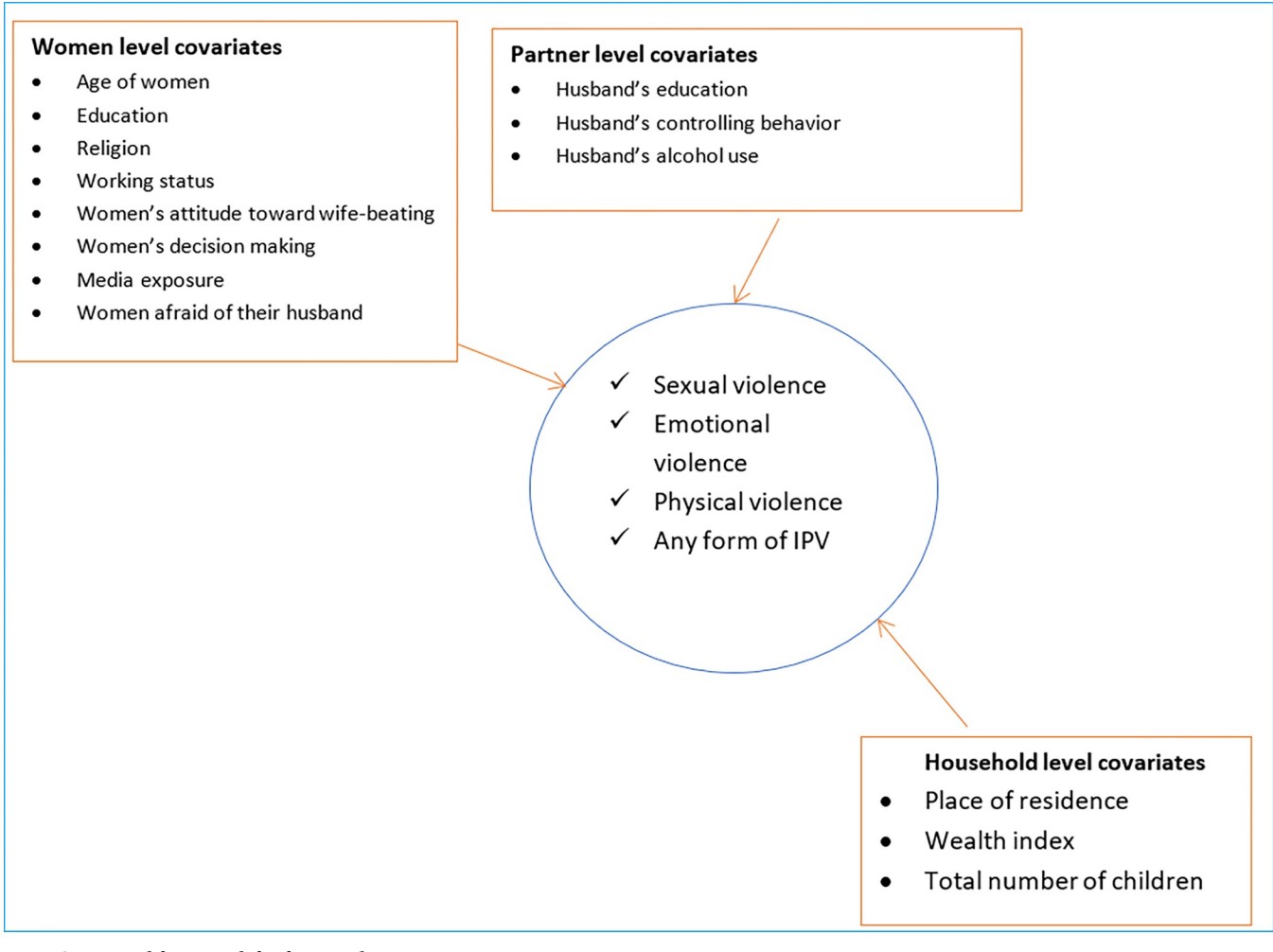

**Fig 2. Conceptual framework for features description.**

mixed model (GLMM) [42–46] to examine the effect of the women, husband, and household characteristics on IVP measures for married women 15–49 age in Ethiopia. The adopted GLMM model is:

$$g(\mu_{ij}) = logit\left(\mu_{ij}\right) = \log\left(\frac{\mu_{ij}}{1-\mu_{ij}}\right) = \log\left(\frac{P(y_{ij}=1)}{P(y_{ij}=0)}\right) = \eta_{ij},$$

where $\eta_{ij} = \beta_0 + \beta_1 x_{1ij} + \cdots + \beta_k x_{kij} + u_{0j}$, $X_{1ij}$, through $X_{kij}$, note the k explanatory variables measured on women, husbands and households. The $\mu_{ij}$ and $1$-$\mu_{ij}$ are respectively the probability of a women experiencing IPV and not experiencing IPV (j = 1,..., 72 zones, i = 1,..., $n_j$ women within each zone): where $\beta_0$ is the log odds of intercept; $\beta_1 \ldots \beta_k$ are effect sizes of women and household-level covariates $u_{0j}$ are random errors at Zone level. The distribution of $u_{0j} \sim N(0, \sigma_{u0}^2)$. The intra-class correlation (ICC) was computed using between-Zone variance and within the Zone, variance (ICC = $\left(\sigma u^2 / \sigma_u^2 + \sigma_e^2\right)$ [47–49].

### Ethical consideration

This study used datasets of national representative demographic health surveys. Therefore, ethical is approval not required. But, datasets for this study were requested by providing a clear explanation about the objectives and necessity of this study. We registered and requested the DHS dataset to the online database (www.dhsprogram.com) and received an authorization letter to download the requested datasets.

## Results

The result in Fig 3 revealed the experience of different forms of intimate partner violence, single or in combinations among women who reported experience of one or more forms of IPV. The result indicated that 1,053 (22.31%), 962 (20.38%) and 358 (7.58%) of women reported that they have experienced emotional, physical and sexual intimate partner violence alone respectively. However, 658 (13.94%) of the married women had experienced both physical and emotional violence and 204 (4.32%) of women who had experienced all forms of intimate partner violence. Besides, the result shows that 1,379 (30.35%) of women who had experienced any forms of intimate partner violence.

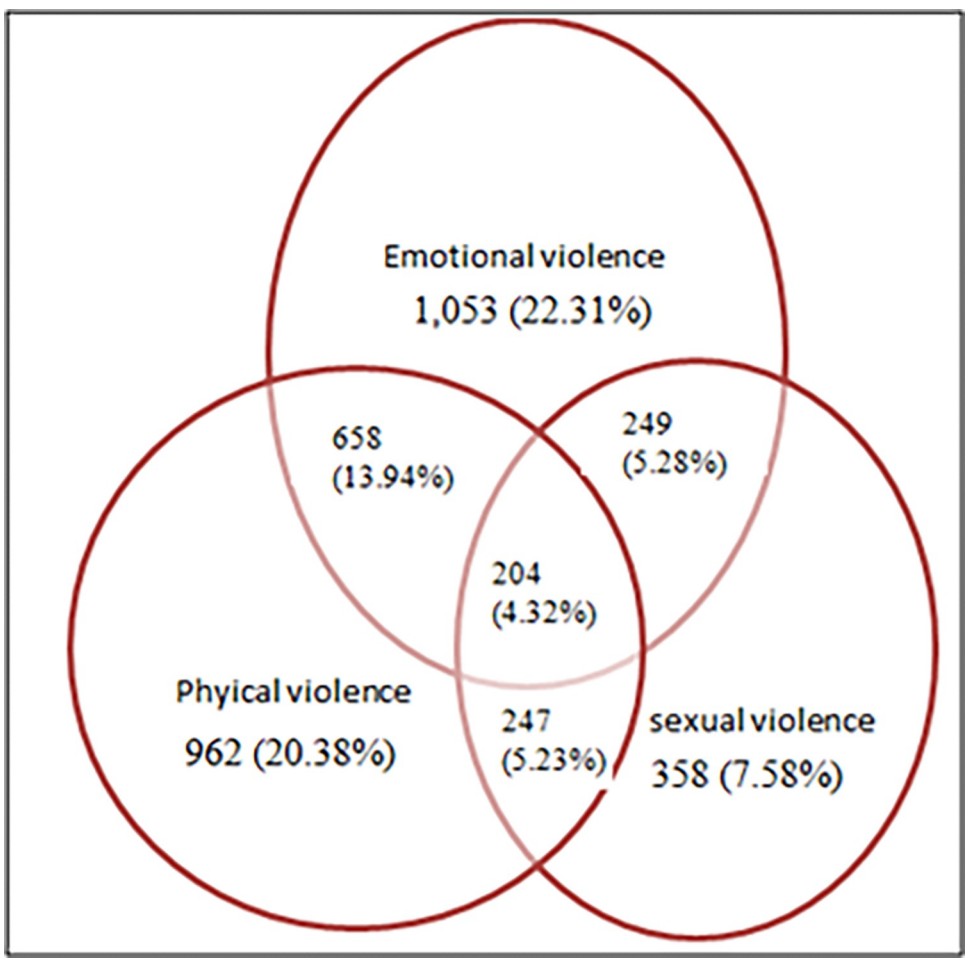

**Fig 3. Overlap prevalence revealing frequencies and percentage of physical, emotional and sexual partner violence against married women in Ethiopia.**

## Spatial results

The proportion of IPV varied a lot between the administrative zones in Ethiopia. Shinile in Somali region was the zone with most physical IPV, while women in North Shewa and most of the zones in SNNP regions (Amaro, Basketo, Konso and Yem special woredas) administrative zones reported lowest Physical IPV prevalence. Generally, there was a zonal variations in the proportion of physical, sexual, emotional, and any form of IPV among the administrative zones in Ethiopia (Fig 4).

The analysis included 4,720 eligible married women. The descriptive statistics was done to present the proportion of socio-demographic characteristics and women who had experienced any of the IPV for each category in the covariates. More than half of all married women had no formal education 2,735 (57.94) and 1,953 (47.37%) of their husbands had no education. Almost 37% (n = 1,716) of women were employed and almost 21% of the women were not empowered in household decision-making activities. Two thousand seventy-eight (44.03%) of the married women never afraid of their husbands. The prevalence of physical, sexual, and emotional violence among married women was 20.38%, 7.58%, and 22.31%, respectively. The least and most prevalent from IPV were sexual (7.58%) and emotional IPV (22.31%). One in every three (30.15%) married women experienced any form of IPV in their lifetime. Twenty percent, 8.37%, 23.33%, and 31% of women with no formal education respectively experienced physical, sexual, and emotional and any form of IPV and the prevalence decreased with increased values of education. Experience of any form of IPV was highest among the oldest age group, but lowest among married women aged 15–24 years. The prevalence of the sexual, physical, emotional, and any form of IPV was lower among women living in the urban areas, and those belong to the higher wealth quintiles. Married women with controlling behavior

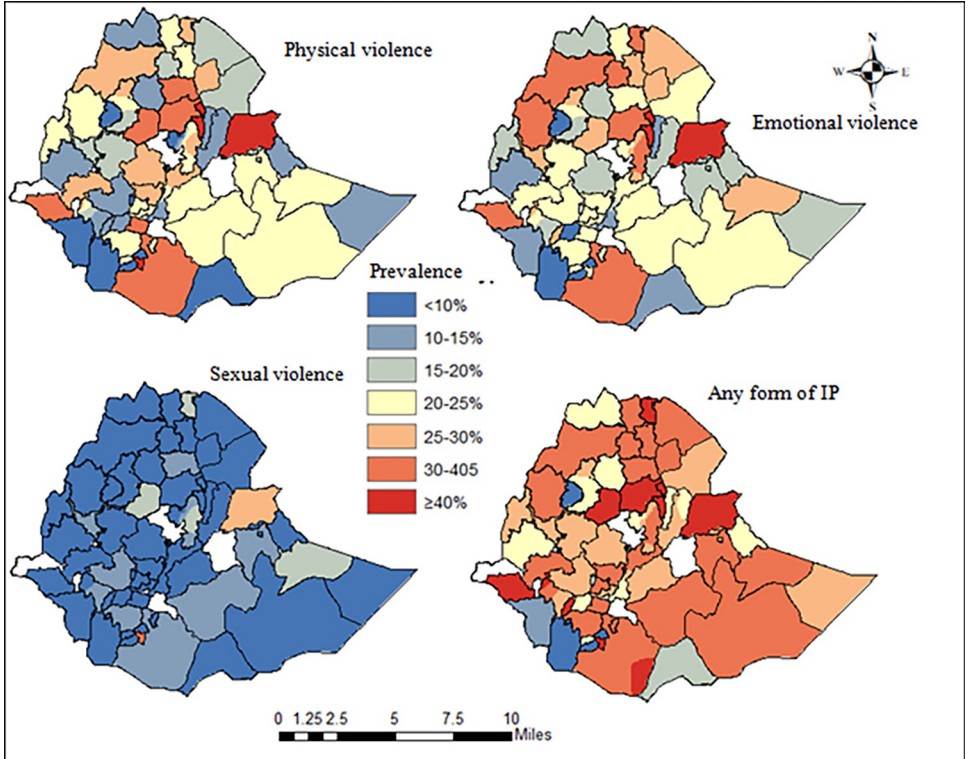

**Fig 4. The prevalence of different types of IPV among the 72 administrative zones in Ethiopia.**

from their partner/spouse, women with no decision making participation, women without media exposure, husband's alcohol use, and women afraid of their spouse, had a higher prevalence of all forms of IPV. A statistically strong relationship between decision-making in the household and experience of any form of IPV was demonstrated. Women who reported not involving in decision making had a higher odds of experiencing IPV compared to those who usually made decision. The highest prevalence of IPV (46.55%) was observed among women who had a husband who was a drinker compared to those who had no drinker husband (24.44%). Similarly, the IPV prevalence varies according to the husband's education level. The highest IPV percentage among women was observed whose husband's a primary education (33.18%) had compared with those who had no education (28.93%) and higher education (24.54%) (Table 2).

Table 3 presents the results of the mixed effect logistic regression analysis with 95% level of confidence, which shows the association of fixed effects and the random effects (zones) for ever experience of physical, sexual, and emotional IPV among married women aged 15–49 years. We found a significant association between the ages of married women with emotional and any form of IPV. Compared to married women aged 15–24 years, women aged 35–44 60% (AOR = 1.60; 95% CI 1.17, 2.19) and 45% (AOR = 1.45; 95% CI 1.09, 1.64) were more likely to experience emotional and at least one form of IPV respectively. The educational status of women had an impact on emotional violence only. Women with higher education are 40% less likely to experience emotional violence compared with no formal education. Women with decision-making autonomy in the household were 27% (AOR 0.73, 95% CI: 0.55, 0.98) and 25% (AOR 0.75: 95% CI: 0.58, 0.97) less likely to report experiencing physical and emotional violence respectively. More importantly, compared to women who were never afraid of their husbands, women who were most of the time afraid of their husbands were more likely to report experiencing physical, sexual, and emotional partner violence. Particularly, women who afraid of their husband sometimes and most of the time were about 3 times (AOR 2.85, 95% CI: 2.37, 3.43) and 5 times (AOR 4.94, 95% CI: 3.94, 6.20) more likely to report experiencing of IPV respectively. Women married to a partner who drank alcohol, were about 4 times (AOR 3.58, 95% CI: 2.82, 4.55), 3 times (AOR 1.81, 3.48), and 3 times (AOR 2.57, 95% CI: 2.06, 3.22) more likely to report experiencing physical, sexual and emotional intimate partner violence respectively (Table 3).

The best linear unbiased predictor (BLUP) shows that there are variations of IPV (physical, emotional, and sexual) among the administrative zones of Ethiopia. Based on the BLUP for the zone-level random effect, the zones were ranked and the best five (those with lowest standardized BLUP values) and top 5 "worst" (those with the highest standardized BLUP values) performing zones in terms of IPV were identified. Hence, women who live in Metekel zone in Benshangul gumz region, Guraghe zone in SNNP, Dege Habur zone in Somali region, South wollo and North Gondar in Amhara region had a higher IPV. However, women who lived in Western Tigray, Jimma and East shewa in Oromia region, Jigjiga in Somali region, and Zone 3 in Addis Ababa had the lowest IPV (Table 4).

## Discussion

This study aimed to examine the prevalence of intimate partner violence among Ethiopian administrative zones and identify the determinant factors using data from the 2016 EDHS and shapefiles. Women in Ethiopia and around the globe experience different forms of violence (such as physical, emotional, and sexual) throughout their lives. However, a single intimate violence index such as sexual, emotional, or physical does not show the holistic picture of violations among married women. To alleviate this problem, we adopted a multifaceted single

**Table 2. The prevalence of partner violence by socio-demographic characteristics among ever-partnered or married 15–49 year old women in Ethiopia (n = 4720).**

| Variables | Sample, n (%) | Physical, n (%) | Sexual, n (%) | Emotional, n(%) | IPV, n (%) |
|---|---|---|---|---|---|
| Outcome variables | | 962 (20.38) | 358 (7.58) | 1,053 (22.31) | 1,423 (30.15) |
| **Women level covariates** | | | | | |
| age in 5-year groups | | | | | |
| 15–24 | 1,140 (24.15) | 201(17.63) | 67 (5.88) | 208 (18.25) | 295 (25.88) |
| 25–34 | 2,012 (42.63) | 419 (20.83) | 158 (7.85) | 434 (21.57) | 610 (30.32) |
| 35–44 | 1,219 (25.83) | 263 (21.58) | 102 (8.37) | 305 (25.02) | 391 (32.08) |
| 45–49 | 349 (7.39) | 79 (22.64) | 31 (8.88) | 106 (30.37) | 127 (36.39) |
| Education | | | | | |
| No Education | 2,735 (57.94) | 556 (20.33) | 229 (8.37) | 638 (23.33) | 815 (30.99) |
| Primary | 1,320 (27.97) | 288 (21.82) | 100 (7.58) | 304 (23.03) | 399 (31.42) |
| Secondary | 433 (9.17) | 86 (19.86) | 21 (4.85) | 78 (18.01) | 117 (27.79) |
| Higher | 232 (4.92) | 32 (13.79) | 8 (3.45) | 33 (14.22) | 48 (21.52) |
| Religion | | | | | |
| Orthodox | 1,805 (38.24) | 391 (21.66) | 166 (9.20) | 437 (24.21) | 589 (32.63) |
| Muslim | 1,993 (42.22) | 345 (17.31) | 116 (5.82) | 374 (18.77) | 510 (25.59) |
| Protestant | 835 (17.69) | 197 (23.59) | 69 (8.26) | 210 (25.15) | 285 (34.13) |
| Traditional And Others | 87 (1.84) | 29 (33.33) | 7 (8.05) | 32 (36.78) | 39 (44.83) |
| Working status | | | | | |
| No | 3,004 (63.64) | 560 (18.64) | 232 (7.72) | 638 (21.24) | 856 (28.50) |
| Yes | 1,716 (36.36) | 402 (23.43) | 126 (7.34) | 415 (24.18) | 567 (33.04) |
| women's attitude to wife-beating | | | | | |
| Does not justify wife-beating | 136 (2.880 | 32 (23.53) | 12 (8.82) | 30 (22.06) | 4 (33.09) |
| Justifies wife-beating | 4,584 (97.12) | 930 (20.29) | 346 (7.55) | 1,023 (22.32) | 1,378 (30.06) |
| women decision making | | | | | |
| no participation | 972 (20.59) | 280 (28.81) | 105 (10.80) | 274 (28.19) | 363 (37.35) |
| moderate participation | 3,225 (68.33) | 561 (17.40) | 213 (6.60) | 652 (20.22) | 894 (27.72) |
| high level | 523 (11.08) | 121 (23.14) | 40 (7.65) | 127 (24.28) | 166 (31.74) |
| media exposure | | | | | |
| no | 2,865 (60.70) | 588 (20.52) | 246 (8.59) | 663 (23.14) | 878 (30.65) |
| yes | 1,855 (39.30) | 374 (20.16) | 112 (6.04) | 390 (21.02) | 545 (29.38) |
| women afraid husband | | | | | |
| never afraid | 2,078 (44.03) | 215 (10.35) | 76 (3.66) | 229 (11.02) | 348 (16.75) |
| most of the time | 783 (16.59) | 336 (42.91) | 136 (17.37) | 340 (43.44) | 434 (55.43) |
| sometimes | 1,859 (39.39) | 411 (22.11) | 146 (7.85) | 484 (26.04) | 641 (34.48) |
| **Partner level covariates** | | | | | |
| husband's Education | | | | | |
| No Education | 1,953 (47.37) | 361 (18.48) | 143 (7.32) | 429 (21.97) | 565 (28.93) |
| Primary | 1,287 (31.22) | 279 (21.68) | 106 (8.24) | 317 (24.63) | 427 (33.18) |
| Secondary | 493 (11.96) | 86 (17.44) | 27 (5.48) | 88 (17.85) | 121 (24.54) |
| Higher | 390 (9.46) | 51 (13.08) | 11 (2.82) | 50 (12.82) | 77 (19.74) |
| Partner's controlling behavior | | | | | |
| No controlling behavior | 2,236 (47.37) | 175 (7.83) | 64 (2.86) | 190 (8.5) | 297 (13.28) |
| Has controlling behavior | 2,484 (52.63) | 787 (31.68) | 294 (11.84) | 863 (34.74) | 1,126 (45.33) |
| husband alcohol use | | | | | |
| no | 3,502 (74.19) | 546 (15.59) | 198 (5.65) | 633 (18.08) | 856 (24.44) |
| yes | 1,218 (25.81) | 416 (34.15) | 160 (13.14) | 420 (34.48) | 567 (46.55) |
| **household level covariates** | | | | | |

*(Continued)*

**Table 2.** (Continued)

| Variables | Sample, n (%) | Physical, n (%) | Sexual, n (%) | Emotional, n(%) | IPV, n (%) |
|---|---|---|---|---|---|
| **Outcome variables** | | **962 (20.38)** | **358 (7.58)** | **1,053 (22.31)** | **1,423 (30.15)** |
| Place of residence | | | | | |
| Rural | 3,509 (74.34) | 731(20.83) | 295 (8.41) | 798 (22.74) | 1,088 (31.01) |
| Urban | 1,211 (25.66) | 231 (19.08) | 63 (5.20) | 255 (21.06) | 335 (27.66) |
| Wealth Index | | | | | |
| Poorest | 1,412 (29.92) | 261 (18.48) | 109 (7.72) | 292 (20.68) | 400 (28.33) |
| Poorer | 729 (15.44) | 159 (21.81) | 72 (9.88) | 174 (24.14) | 234 (32.10) |
| Middle | 660 (13.98) | 147 (22.27) | 63 (9.55) | 166 (25.15) | 229 (34.70) |
| Richer | 610 (12.92) | 149 (24.43) | 54 (8.85) | 159 (26.07) | 208 (34.10) |
| Richest | 1,309 (27.73) | 246 (18.79) | 60 (4.58) | 260 (19.86) | 352 (26.89) |
| total number of children | | | | | |
| 0 | 471 (9.98) | 70 (14.86) | 26 (5.52) | 76 (14.14) | 102 (21.66) |
| 1–4 | 2,130 (45.13) | 450 (21.13) | 145 (6.81) | 451 (21.17) | 642 (30.14) |
| 4 or more | 2,119 (44.89) | 442 (20.86) | 187 (8.82) | 526 (24.82) | 679 (32.04) |

index known as intimate partner violence. Using this index, we investigated the disparities of Ethiopian married women's violation status in space which shows the administrative zones, the second level of the Ethiopian administrative level where the social service delivery decision-making process is made.

Most of the studies conducted in Ethiopia had only reported geographical variations of IPV at higher (country/region) aggregated levels [17, 19, 21, 22, 50], and zonal level variation is rarely examined. A closer look into the contents of the studies shows that their IPV data is masked in higher-level geographical aggregates, and hurt lower levels (zones in this context). This is inconsistent with the decentralized system of governance in Ethiopia. The zone is the administrative level where operation planning, resource allocation, and implementation of health services (including women empowerment for decision-making) are made. Hence identifying the problem of IPV and its variation among administrative zones would provide deeper insight into the country's health priorities of women in the population. Particularly, this would help Zonal health departments to make informed decisions and actions in their planning, follow-up, monitoring, and evaluation of women's empowerment and decision-making ability at lower levels.

We found that IPV is a considerable public health problem and the prevalence was 30.15 indicating that one in every three women had experienced at least one form of IPV. The overall prevalence was higher than studies conducted in Saudi Arabia and Benin [51–53], but lower than studies conducted in sub-Saharan Africa, Zimbabwe, Gambia, and studies in 46 low-middle income countries (LMICs) [11, 54–56]. The presence of all spousal violence varied considerably across the administrative zones of the country. At the women level covariates, older age, no formal education level, no media experience, working status and afraid of their partner were positively associated with some forms of IPV. Women of higher age were more likely to experience emotional and IPV. However, other researchers found that the risk of experiencing IPV was higher among the lower age groups [57–59]. The possible reasons for the contradictory findings could be cultural and area level differences between the study samples and IPV. Women married to a spouse who drank alcohol had increased the odds of physical, sexual, emotional and IPV [57, 58, 60]. This is due to the strong impact of unlimited alcohol intake which may change the behavior of husbands. Partner's controlling behavior in a relationship was found to be a protective factor against physical, sexual, emotional and IPV.

**Table 3. Factors associated with experiencing physical, emotional, sexual and intimate partner violence using GLMM.**

| Variables | Physical | Sexual | Emotional | Intimate partner (IPV) |
|---|---|---|---|---|
| **women level covariates** | AORs 95% CI | AORs 95% CI | AORs 95% CI | AORs 95% CI |
| age in 5-year groups | | | | |
| 15–24 | 1 | 1 | 1 | 1 |
| 25–34 | 1.04 (0.80, 1.36) | 1.14 (0.75, 1.73) | 1.10 (0.84, 1.41) | 1.13 (0.89, 1.42) |
| 35–44 | 1.32 (0.96, 1.83) | 1.30 (0.80, 2.14) | 1.60 (1.17, 2.19)*** | 1.45 (1.09, 1.64)** |
| 45–49 | 0.99 (0.64, 1.54) | 1.25 (0.66, 2.37) | 1.72 (1.15, 2.58)** | 1.44 (0.99, 2.11) |
| Education | | | | |
| No Education | | | 1 | 1 |
| Primary | | | 1.02 (0.83, 1.26) | 1.18 (0.95, 1.46) |
| Secondary | | | 0.77 (0.55, 1.09) | 1.44 (0.99, 1.08) |
| Higher | | | 0.60 (0.39, 0.62)** | 1.12 (0.67, 1.87) |
| Religion | | | | |
| Orthodox | 1 | | 1 | 1 |
| Muslim | 1.62 (1.24, 2.12)*** | | 1.16 (0.91, 1.49) | 1.32 (1.05, 1.66)* |
| Protestant | 1.52 (1.15, 2.01)*** | | 1.22 (0.94, 1.60) | 1.40 (1.09, 1.79)** |
| Traditional And Others | 2.31 (1.29, 4.15)*** | | 2.20 (1.25, 3.88)** | 2.40 (1.38, 4.15)** |
| Respondent Currently Working | | | | |
| No | 1 | 1 | 1 | 1 |
| Yes | 1.28 (1.05, 1.55)** | 0.81 (0.60, 1.08) | 1.13 (0.93, 1.36) | 1.17 (0.98, 1.39) |
| women decision making | | | | |
| No | 1 | | 1 | |
| Yes | 0.73 (0.55, 0.98)* | | 0.75 (0.58, 0.97)** | |
| media exposure | | | | |
| no | | | 1 | |
| yes | | | 0.92 (0.74, 1.15) | |
| women afraid husband | | | | |
| never afraid | 1 | 1 | 1 | 1 |
| most of the time | 4.97 (3.87, 6.39)*** | 4.23 (2.93, 6.10)*** | 4.65 (3.65, 5.93)*** | 4.94 (3.94, 6.20)*** |
| sometimes | 2.40 (1.93, 2.98)*** | 2.32 (1.65, 3.26)*** | 2.90 (2.36, 3.57)*** | 2.85 (2.37, 3.43)*** |
| **Partner level covariates** | | | | |
| husband's education | | | | |
| No Education | 1 | 1 | 1 | 1 |
| Primary | 1.03 (0.82, 1.28) | 1.11 (0.81, 1.51) | 1.03 (0.83, 1.26) | 1.03 (0.85, 1.25) |
| Secondary | 0.91 (0.64, 1.29) | 1.21 (0.72, 2.04) | 0.77 (0.55, 1.09) | 0.75 (0.55, 1.03) |
| Higher | 0.79 (0.51, 1.24) | 0.70 (0.33, 1.52) | 0.600 (0.39, 0.92)** | 0.67 (0.45, 0.98)* |
| Partner's controlling behavior (MC) | | | | |
| No controlling behavior | 1 | 1 | 1 | 1 |
| Has controlling behavior | 4.06 (3.32, 4.97)*** | 3.38 (2.47, 4.64)*** | 4.78 (3.93, 5.79)*** | 4.43 (3.75, 5.25)*** |
| husband alcohol use | | | | |
| no | 1 | 1 | 1 | |
| yes | 3.58 (2.82, 4.55)*** | 2.51 (1.81, 3.48)*** | 2.57 (2.06, 3.22*** | 3.30 (2.66, 4.09)*** |
| **household level covariates** | | | | |
| Place of residence | | | | |
| Rural | | | 1 | |
| Urban | | | 0.75 (0.52, 1.08) | |
| Wealth Index | | | | |
| Poorest | 1 | 1 | | 1 |

(*Continued*)

**Table 3.** (Continued)

| Variables | Physical | Sexual | Emotional | Intimate partner (IPV) |
|---|---|---|---|---|
| Poorer | 1.03 (0.78, 1.35) | 1.12 (0.78, 1.61) | | 0.94 (0.74, 1.20) |
| Middle | 1.03 (0.77, 1.37) | 1.03 (0.70, 1.52) | | 1.08 (0.84, 1.39) |
| Richer | 0.94 (0.96, 1.27) | 0.76 (0.49, 1.18) | | 0.78 (0.60, 1.03) |
| Richest | 0.95 (0.62, 1.45) | 0.45 (0.23, 0.89) | | 0.77 (0.53, 1.12) |
| total number of children | | | | |
| 0 | 1 | | 1 | 1 |
| 1–3 | 1.35 (0.91, 2.01) | | 1.21 (0.83, 1.76) | 1.32 (0.94, 1.84) |
| 4 and more | 1.25 (0.81, 1.96) | | 1.24 (0.81, 1.91) | 1.28 (0.87, 1.88) |
| Random Effects components | | | | |
| ICC (95% CI) | 0.17 (0.13, 0.22)*** | 0.24 (0.18, 0.33)*** | 0.13 (0.09, 0.17)*** | 0.16 (0.13, 0.20)*** |

GLMM: generalized linear mixed effect model, aOR = adjusted odds ratio, CI = confidence interval

[c] P < 0.a0001

[b] P < 0.001

[a] P < 0.05

Moreover, the spatial heterogeneity and inequality of IPV was analyzed and mapped at the second administrative levels in Ethiopia. The presence of more than 80 ethnic groups in Ethiopia's 72 zones means that cultural practices and gender norms differ. The zones with the highest IPV status and the highest degree of inequality were identified. It is already known that IPV prevalence differs among geographic areas. Many studies limitation have been conducted in Ethiopia, but most of them were focused in only regions or states. There is clearly a variation of IPV at the second administrative zones even within the same regions, hence, we focused on the second level administrative areas (zones).

The study has both strengths and limitations. This large dataset made it possible to apply the high-level identify the important factors. However, this study has some limitations. Firstly, we considered only one recent DHS dataset, and hence we did not model the variables over time. Secondly, the data is cross-sectional so we can only make conclusions on statistical association (not causality). Thirdly, the research could be susceptible to social desirability and potential recall bias due to its reliance on self-reported data.

## Conclusions

This study aimed to investigate the magnitude of intimate partner violence and its influencing factors by utilizing national EDHS data, as part of the crucial efforts to accomplish Sustainable Development Goals (SDGs) related to gender equality and women's empowerment, ultimately contributing to ending violence against women. The spatial units for intimate partner violence analysis is "zones" (n = 72 zones in Ethiopia). The result showed that the proportion of women who had experienced physical, sexual, emotional or at least one form of this IPV was high in Ethiopia. Our findings suggest there are zone-wise variations in the prevalence of IPV. Taking into account the context and cultural norms among zones in Ethiopia, the IPV cases could even be under-reported. Our finding shows where and towards which populations IPV resources should be allocated could help national health policymakers develop appropriate sets of interventions or prevent the use of incorrect interventions for intimate partner violence control and prevention in Ethiopia. Being younger age, women's decision-making autonomy, never being afraid of their husbands, no husband's alcohol drink, partners without controlling behavior were negatively associated with any form of IPV. Hence improving the power of

**Table 4. Model based comparisons of different intimate partner violence among administrative zones in Ethiopia in married women.**

| Regions of Ethiopia | Districts/ zones | Physical | | Emotional | | Sexual | | IPV | |
|---|---|---|---|---|---|---|---|---|---|
| | | BLUP | Ranking | BLUP | Ranking | BLUP | Ranking | BLUP | Ranking |
| Addis Ababa | AAUnknown | 0.0883 | 64 | 0.0068 | 48 | 0.0910 | 64 | 0.1951 | 64 |
| | AAZone 1 | 0.0435 | 54 | 0.0106 | 49 | 0.0436 | 53 | 0.1068 | 54 |
| | AAZone 2 | -0.0958 | 9 | -0.0350 | 11 | -0.0937 | 8 | -0.2155 | 7 |
| | AAZone 3 | -0.1137 | 6 | -0.0435 | 3 | -0.1124 | 5 | -0.2606 | 5 |
| | AAZone 4 | 0.0107 | 47 | 0.0237 | 63 | 0.0062 | 44 | 0.0496 | 51 |
| | AAZone 5 | 0.0019 | 44 | -0.0050 | 31 | 0.0063 | 45 | 0.0123 | 41 |
| | AAZone 6 | -0.0730 | 13 | -0.0416 | 4 | -0.0697 | 13 | -0.1753 | 11 |
| Afar | Zone 1 | -0.0603 | 17 | 0.0207 | 61 | -0.0527 | 18 | -0.0833 | 20 |
| Afar | Zone 2 | **-0.1202** | **4** | 0.0476 | 67 | -0.1238 | 3 | -0.1874 | 8 |
| Afar | Zone 3 | -0.0962 | 8 | -0.0076 | 28 | -0.0917 | 9 | -0.1866 | 9 |
| Afar | Zone 4 | 0.0759 | 59 | 0.0129 | 52 | 0.0745 | 58 | 0.1723 | 59 |
| Afar | Zone 5 | -0.0147 | 34 | -0.0095 | 25 | -0.0142 | 33 | -0.0294 | 33 |
| Amhara | Awi | -0.0379 | 23 | -0.0155 | 22 | -0.0373 | 24 | -0.0816 | 21 |
| | Bar Dar Sp. Zone | 0.0157 | 49 | 0.0029 | 44 | 0.0154 | 48 | 0.0430 | 49 |
| | East Gojam | 0.1355 | 67 | 0.0179 | 59 | 0.1329 | 66 | 0.2952 | 67 |
| | North Gonder | 0.1663 | 68 | 0.0814 | 70 | 0.1546 | 67 | 0.4113 | 69 |
| | North Shewa (K3) | -0.0137 | 35 | 0.0287 | 65 | -0.0133 | 34 | 0.0107 | 40 |
| | North Wollo | 0.2085 | 71 | 0.0023 | 43 | 0.2048 | 70 | 0.4246 | 70 |
| | Oromia Zone | -0.0100 | 36 | -0.0037 | 34 | -0.0098 | 36 | -0.0145 | 34 |
| | South Gonder | -0.0764 | 12 | -0.0177 | 19 | -0.0713 | 12 | -0.1564 | 14 |
| | South Wollo | 0.1811 | 70 | 0.0537 | 68 | 0.1816 | 69 | 0.4254 | 71 |
| | Wag Hemira | 0.0201 | 51 | 0.0234 | 62 | 0.0209 | 50 | 0.0734 | 52 |
| | West Gojam | -0.0632 | 16 | -0.0396 | 6 | -0.0643 | 15 | -0.1581 | 13 |
| Benshangul-Gumaz | Asosa | -0.0150 | 33 | -0.0621 | 1 | -0.0190 | 31 | -0.0871 | 19 |
| | Kemashi | -0.0023 | 40 | 0.0174 | 58 | -0.0030 | 40 | 0.0211 | 44 |
| | Metekel | 0.0640 | 57 | 0.1115 | 72 | 0.0549 | 54 | 0.2393 | 66 |
| Dire Dawa | Dire Dawa | -0.0007 | 42 | -0.0272 | 14 | -0.0120 | 35 | -0.0308 | 32 |
| Gambela Peoples | GZone 1 | 0.0916 | 65 | 0.0276 | 64 | 0.0898 | 63 | 0.2180 | 65 |
| Gambela Peoples | GZone 2 | -0.0394 | 22 | -0.0055 | 30 | -0.0396 | 23 | -0.0756 | 24 |
| Harari People | Hundene | -0.0727 | 14 | -0.0157 | 21 | -0.0654 | 14 | -0.1449 | 16 |
| Oromia | Arsi | -0.0342 | 25 | -0.0086 | 26 | -0.0295 | 27 | -0.0633 | 27 |
| | Bale | -0.0049 | 38 | -0.0171 | 20 | 0.0006 | 42 | -0.0123 | 35 |
| | Borena | 0.0174 | 50 | -0.0045 | 33 | 0.0188 | 49 | 0.0407 | 48 |
| | East Harerghe | 0.0345 | 52 | -0.0332 | 13 | 0.0386 | 52 | 0.0489 | 50 |
| | East Shewa | -0.1145 | 5 | -0.0562 | 2 | -0.1134 | 4 | -0.2751 | 3 |
| | East Wellega | -0.0039 | 39 | 0.0140 | 53 | -0.0024 | 41 | 0.0166 | 43 |
| | Illubabor | 0.0803 | 62 | 0.0050 | 45 | 0.0788 | 60 | 0.1731 | 60 |
| | Jimma | -0.1485 | 2 | -0.0208 | 17 | -0.1422 | 2 | -0.3025 | 2 |
| | North Shewa (K4) | -0.0912 | 10 | 0.0114 | 50 | -0.0875 | 10 | -0.1582 | 12 |
| | West Harerghe | -0.0637 | 15 | -0.0390 | 9 | -0.0596 | 16 | -0.1533 | 15 |
| | West Shewa | 0.0599 | 56 | -0.0347 | 12 | 0.0553 | 55 | 0.0894 | 53 |
| | West Wellega | -0.0245 | 29 | -0.0203 | 18 | -0.0214 | 30 | -0.0572 | 29 |
| Somali | Dege Habur | 0.0403 | 53 | 0.0603 | 69 | 0.0330 | 51 | 0.1426 | 55 |
| | Jijiga | -0.1248 | 3 | -0.0411 | 5 | -0.1065 | 6 | -0.2634 | 4 |
| | Liben | -0.1058 | 7 | -0.0393 | 7 | -0.1039 | 7 | -0.2401 | 6 |
| | Shinile | 0.0652 | 58 | 0.0061 | 46 | 0.0648 | 57 | 0.1452 | 57 |

*(Continued)*

**Table 4.** (Continued)

| Regions of Ethiopia | Districts/ zones | Physical | | Emotional | | Sexual | | IPV | |
|---|---|---|---|---|---|---|---|---|---|
| | | BLUP | Ranking | BLUP | Ranking | BLUP | Ranking | BLUP | Ranking |
| | Unknown | 0.1103 | 66 | -0.0378 | 10 | 0.1109 | 65 | 0.1924 | 63 |
| | Welwel & Warder | -0.0486 | 19 | -0.0046 | 32 | -0.0530 | 17 | -0.0971 | 18 |
| SNNP | Amaro Special Woreda | -0.0158 | 32 | 0.0063 | 47 | 0.124 | 72 | -0.0005 | 38 |
| | Basketo Special Woreda | -0.0011 | 41 | -0.0033 | 37 | -0.0093 | 38 | -0.0047 | 37 |
| | Benchi Maji | -0.0331 | 26 | -0.0112 | 24 | -0.0311 | 25 | -0.0665 | 26 |
| | Burji Special Woreda | -0.0536 | 18 | -0.0392 | 8 | -0.0514 | 19 | -0.1352 | 17 |
| | Dawuro | -0.0818 | 11 | -0.0216 | 16 | -0.0837 | 11 | -0.1782 | 10 |
| | Derashe Special Woreda | -0.0006 | 43 | -0.0013 | 39 | 0.0036 | 43 | 0.0107 | 39 |
| | Gamo Gofa | 0.0596 | 55 | 0.0192 | 60 | 0.0563 | 56 | 0.1442 | 56 |
| | Gedeo | -0.0177 | 31 | 0.0173 | 57 | -0.0142 | 32 | -0.0056 | 36 |
| | Guraghe | 0.3133 | 72 | 0.1087 | 71 | 0.3045 | 71 | 0.7354 | 72 |
| | Hadiya | 0.0790 | 60 | 0.0145 | 55 | 0.0816 | 61 | 0.1840 | 62 |
| | Kaffa | -0.0452 | 20 | -0.0036 | 36 | -0.0402 | 22 | -0.0800 | 22 |
| | Kembata Alaba Tembaro | 0.0801 | 61 | 0.0147 | 56 | 0.0782 | 59 | 0.1820 | 61 |
| | Konso Special Woreda | 0.0134 | 48 | -0.0021 | 38 | -0.0068 | 39 | 0.0135 | 42 |
| | Konta Special Woreda | 0.0069 | 45 | 0.0120 | 51 | 0.0077 | 46 | 0.0356 | 46 |
| | Shaka | 0.0093 | 46 | -0.0037 | 35 | 0.0083 | 47 | 0.0229 | 45 |
| | Sidama | 0.0851 | 63 | -0.0070 | 29 | 0.0841 | 62 | 0.1713 | 58 |
| | South Omo | -0.0244 | 30 | -0.0231 | 15 | -0.0247 | 29 | -0.0632 | 28 |
| | Wolayita | 0.1672 | 69 | -0.0078 | 27 | 0.1589 | 68 | 0.3273 | 68 |
| | Yem Special Woreda | -0.0290 | 28 | 0.0005 | 42 | -0.0281 | 28 | -0.0476 | 30 |
| Tigray | Central Tigray | -0.0373 | 24 | -0.0006 | 41 | -0.0405 | 21 | -0.0694 | 25 |
| | Easetern Tigray | -0.0326 | 27 | 0.0143 | 54 | -0.0301 | 26 | -0.0394 | 31 |
| | Mekele | -0.0414 | 21 | -0.0009 | 40 | -0.0438 | 20 | -0.0772 | 23 |
| | Southern Tigray | -0.0052 | 37 | 0.0421 | 66 | -0.0094 | 37 | 0.0365 | 47 |
| | Western Tigray | -0.1514 | 1 | -0.0112 | 23 | -0.1536 | 1 | -0.3072 | 1 |

women's decision-making autonomy, Precision public health approaches are important for targeting health policies to zones most affected by IPV. Moreover, the best-worst performing zones were identified and this study recommends further investigation into these zones which didn't show any progress in improving intimate partner violence among married women in Ethiopia.

## Acknowledgments

The datasets used in this study were obtained from the DHS program thanks to the authorization received to download the dataset on the website.

## Author Contributions

**Conceptualization:** Meseret Tadesse Fetene, Senait Cherie Adgeh, Haile Mekonnen Fenta.

**Data curation:** Meseret Tadesse Fetene, Haile Mekonnen Fenta.

**Formal analysis:** Meseret Tadesse Fetene.

**Methodology:** Senait Cherie Adgeh, Haile Mekonnen Fenta.

**Resources:** Meseret Tadesse Fetene.

**Software:** Meseret Tadesse Fetene, Senait Cherie Adgeh, Haile Mekonnen Fenta.

**Supervision:** Haile Mekonnen Fenta.

**Visualization:** Senait Cherie Adgeh.

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
