## [Decision Letter · Decision Letter 0]

17 Jul 2024

PONE-D-23-29229Spatial Distributions and determinants of intimate partner violence among married women in Ethiopia across administrative zonesPLOS ONE

Dear Dr. Fenta,

Thank you for submitting your manuscript to PLOS ONE. After careful consideration, we feel that it has merit but does not fully meet PLOS ONE’s publication criteria as it currently stands. Therefore, we invite you to submit a revised version of the manuscript that addresses the points raised during the review process.

**Please address the comments raised by the reviewers. **

We look forward to receiving your revised manuscript.

Kind regards,

Meesha Iqbal

Academic Editor

PLOS ONE

- https://doi.org/10.1371/journal.pone.0256726

- http://dx.doi.org/10.1136/bmjgh-2019-002208

In your revision ensure you cite all your sources (including your own works), and quote or rephrase any duplicated text outside the methods section. Further consideration is dependent on these concerns being addressed.

“No”

6. We note that Figures 1 and 3 in your submission contain [map/satellite] images which may be copyrighted. All PLOS content is published under the Creative Commons Attribution License (CC BY 4.0), which means that the manuscript, images, and Supporting Information files will be freely available online, and any third party is permitted to access, download, copy, distribute, and use these materials in any way, even commercially, with proper attribution. For these reasons, we cannot publish previously copyrighted maps or satellite images created using proprietary data, such as Google software (Google Maps, Street View, and Earth). For more information, see our copyright guidelines: http://journals.plos.org/plosone/s/licenses-and-copyright.

1. You may seek permission from the original copyright holder of Figures 1 and 3 to publish the content specifically under the CC BY 4.0 license. 

Reviewers' comments:

Reviewer's Responses to Questions

**Comments to the Author**

1. Is the manuscript technically sound, and do the data support the conclusions?

Reviewer #1: Partly

Reviewer #2: Yes

2. Has the statistical analysis been performed appropriately and rigorously? 

Reviewer #1: Yes

Reviewer #2: Yes

3. Have the authors made all data underlying the findings in their manuscript fully available?

Reviewer #1: Yes

Reviewer #2: Yes

4. Is the manuscript presented in an intelligible fashion and written in standard English?

Reviewer #1: Yes

Reviewer #2: Yes

5. Review Comments to the Author

Reviewer #1: Following Strobe Checklist, these are some comments

1. Introduction:

It is necessary to not only mention that in Ethiopia, the identified prevalence of partner violence varies widely, with physical prevalence ranging from 31% to 76.5%, sexual prevalence from 19.2% to 59%, and emotional prevalence at 51.7%. It is important to specify the social norms and cultural characteristics that are related to this variation and high prevalence. Some of the articles cited by you mention religion, early marriage, and early initiation of sexual relationships as factors, but they are not described in the introduction.

2. A better justification of the research gap is required, to analyzing the data according administrative zones. It is unclear whether these administrative zones encompass different norms, cultures, languages, ethnicities, religions, or socioeconomic levels. It is necessary to describe the actual organization of the population and how there are related to the administrative zones and evaluate whether the analysis should be conducted based on other criteria. If necessary, include this information in the discussion and/or limitations.

3. Specify the study design. Is it a secondary analytical cross-sectional study based on the 2016 Ethiopia Demographic and Health Survey (DHS)?

4. Include general characteristics of the 2016 Ethiopian DHS, such as the complex design being random, two-stage, and balanced. What was the primary sampling unit, the household?

5. There is no information in the manuscript about the population selection criteria for the study (inclusion and exclusion criteria). For example, were women included who were selected for the violence module and were married or cohabiting in the past 12 months? Were women excluded if they did not have privacy when answering violence-related questions? Or if they did not respond to questions about physical, sexual, or emotional violence? Include a flowchart showing the population selection criteria.

6. Statistical power

It is important to describe the statistical power calculation to know if the information on factors associated with intimate partner violence according to administrative areas can include a type II error. It has only been analyzed up to regions, not administrative zones, in other of the articles that you cited that were also secondary analyzes of DHS Ethiopia

7. Specify whether the different types of partner violence were assessed based on the question about violence by the partner in life or in the past 12 months?. Provide more precise operationalization of outcome variables.

8. The independent variables considered in the analysis should be listed based on different studies that are not specified.

9.Statistical analysis

• The level of confidence considered in the statistical analysis is not specified (95%? 99%?)

• It is not specified whether any statistical test, such as the chi-square test with Rao-Scott correction, was used for the bivariate analysis in Table 2. Include the appropriate statistical test if applicable.

• It is not specified what criteria were used to include the adjustment variables in Table 3. Was it based on statistical criteria? If so, specify it. For some forms of violence, different variables were adjusted (e.g., education level for emotional violence, but not for sexual or physical violence). It should be explained in the statistical analysis section whether this was due to multicollinearity or correlation analysis, how it was evaluated, and the criteria used for exclusion.

• Specify the variables used to adjust the comparison model for different types of IPV in the administrative zones.

10. Results.

• Tables.

It is important to mention in the table footer of Table 3 what type of model was used to calculate those models (GLMM?). It would also be helpful to evaluate whether presenting both crude and adjusted models is feasible.

For Table 4, justify whether the power is adequate for comparing zones without a type II error. It may be appropriate to perform an analysis at the regional level. Include in the table footer the specifications used to calculate the analysis, and explain the initials used (BLUP Ranking).

11. Discussion.

It is important to contextualize the findings and discuss whether the geographic delimitation of administrative zones truly reflects the characteristics of partner relationships. Do the 80 ethnic groups in Ethiopia align with this delimitation?

Does partner violence differ within these administrative zones when considering the social determinants model?1 It is possible that some subsets of these zones share common characteristics (social norms, cultural aspects, population structure, religion, socioeconomic level, etc.) and should be analyzed together and contextualized. Additionally, there may be differences within some of these zones.

1Heise L, Greene ME, Opper N, Stavropoulou M, Harper C, Nascimento M, et al. Gender inequality and restrictive gender norms: framing the challenges to health. Lancet. 2019;393(10189):2440-54.

12. It is important to include the study limitations. Some potential limitations could be that the study is based on a cross-sectional design, which only allows for the identification of associations. Additionally, the violence data is self-reported, which may introduce reporting bias due to desirability bias, with individuals more likely to report severe or recent violence,

What happen if there was another person when they asked about intimate partner violence? If there was no privacy, it was recorded in the survey; if so, were they excluded from the analysis?

At Ethiopy DHS the question about violence it is reported who perpetrated the violence. Is it possible that the violence was committed by a former partner and not by his current partner? If there is that possibility, include it in limitations.

Are there any other self-reported variables that may be influenced by desirability bias?

Reviewer #2: Please check the formatting of the manuscript including spacing, punctuation, and grammar.

Methods:

The section should clearly mention that the analysis adjusts for the complex sampling design of the EDHS data by applying sampling weights. It is important to reiterate that all estimates presented are weighted to account for the sampling design

The selection and justification of covariates included in the model are not adequately detailed.

Results:

The interpretation of the spatial distribution results could be more detailed linking them to underlying factors (for example, cultural, socio-economic, regional factors).

Discussion:

While the discussion references existing literature, it could benefit from a more systematic comparison of the study’s findings with those of previous studies in detail. Currently the discussion is brief, but it could be strengthened by a more in-depth analysis of the findings, a clear connection to the Ethiopian context, and should delve into the reasons behind these findings comparing with the previous literature. Furthermore, the strengths and limitations of the study are not clearly outlined.

6. PLOS authors have the option to publish the peer review history of their article (what does this mean?). If published, this will include your full peer review and any attached files.

Reviewer #1: No

Reviewer #2: No

---

## [Author Response · Author response to Decision Letter 0]

24 Jul 2024

In response to comments from Reviewer #1:

Dear reviewer, thank you very much for taking the time to review our manuscript. We seriously considered your comments and suggestions and incorporated them into the revised manuscript. We gave due attention to the typographical errors, results, discussion sections, conclusions, and recommendations so that the manuscript has significantly improved.

Introduction:

1. It is necessary to not only mention that in Ethiopia, the identified prevalence of partner violence varies widely, with physical prevalence ranging from 31% to 76.5%, sexual prevalence from 19.2% to 59%, and emotional prevalence at 51.7%. It is important to specify the social norms and cultural characteristics that are related to this variation and high prevalence. Some of the articles cited by you mention religion, early marriage, and early initiation of sexual relationships as factors, but they are not described in the introduction.

Our response: Thank you for your valuable comment. The main aim of this study is to see the variations of IPV across the administrative zones (where there are cultural and social differences) which is stated (page no. 4, line no 13-18) in the introduction section. Moreover, religion was considered in the analysis and it has a significant impact on the IPV (page 9, Table 3).

2. A better justification of the research gap is required, to analyzing the data according administrative zones. It is unclear whether these administrative zones encompass different norms, cultures, languages, ethnicities, religions, or socioeconomic levels. It is necessary to describe the actual organization of the population and how there are related to the administrative zones and evaluate whether the analysis should be conducted based on other criteria. If necessary, include this information in the discussion and/or limitations.

Our response: Thank you for your comment. We improved the discussion section based on your comment and suggestions (Page 14 , line no. 3- 13). 

3. Specify the study design. Is it a secondary analytical cross-sectional study based on the 2016 Ethiopia Demographic and Health Survey (DHS)?

Our response: Yes, it is, we specified it on page 4, line 21. 

4. Include general characteristics of the 2016 Ethiopian DHS, such as the complex design being random, two-stage, and balanced. What was the primary sampling unit, the household?

Our response: thank you, done (page 4, line 22-31 and page 5: line no. 1-13). 

5. There is no information in the manuscript about the population selection criteria for the study (inclusion and exclusion criteria). For example, were women included who were selected for the violence module and were married or cohabiting in the past 12 months? Were women excluded if they did not have privacy when answering violence-related questions? Or if they did not respond to questions about physical, sexual, or emotional violence? Include a flowchart showing the population selection criteria.

Our response: this is a very good point. We included in the methodology section (page 5, line no. 1-13) 

6. Statistical power: It is important to describe the statistical power calculation to know if the information on factors associated with intimate partner violence according to administrative areas can include a type II error. It has only been analyzed up to regions, not administrative zones, in other of the articles that you cited that were also secondary analyzes of DHS Ethiopia

Our response: thank you. This is applied research and we focused on the administrative zones, instead of regions. We justified the reasons in the previous response. The ICC was computed and added in the revised form of this manuscript with justification (Table 4). 

7. Specify whether the different types of partner violence were assessed based on the question about violence by the partner in life or in the past 12 months?. Provide more precise operationalization of outcome variables.

Our response: thank you for your comment. It is in life time, which was mentioned in the result section, but we added it in the methodology section now (page, 5, line no. 21-23).

8. The independent variables considered in the analysis should be listed based on different studies that are not specified.

Our response: thank you for your suggestions, we added it as a conceptual framework (Fig.2). 

Statistical analysis

9. The level of confidence considered in the statistical analysis is not specified (95%? 99%?)

Our response: it is 95%, it is mentioned in the footnote of Table 4 (p-value <0.05). 

10. It is not specified whether any statistical test, such as the chi-square test with Rao-Scott correction, was used for the bivariate analysis in Table 2. Include the appropriate statistical test if applicable.

Our response: Table 2 is all about the descriptive statistics, which is the distribution of IPV across the covariates. 

11. It is not specified what criteria were used to include the adjustment variables in Table 3. Was it based on statistical criteria? If so, specify it. For some forms of violence, different variables were adjusted (e.g., education level for emotional violence, but not for sexual or physical violence). It should be explained in the statistical analysis section whether this was due to multicollinearity or correlation analysis, how it was evaluated, and the criteria used for exclusion.

Our response: we used crude odds ratio in the context of GLMM and candidate variables for the given IPV form were included in the adjusted odds ratio multivariable cases. We added a sentence in the statistical analysis section now. 

12. Specify the variables used to adjust the comparison model for different types of IPV in the administrative zones. 

Our response: it is different for different IPV forms, but all variables are included in Table 4 and the empty values in the table reveal that the variable is used to adjust the other variables (Table 4). 

Results.

• Tables.

13. It is important to mention in the table footer of Table 3 what type of model was used to calculate those models (GLMM?). It would also be helpful to evaluate whether presenting both crude and adjusted models is feasible.

Our response: thank you, done. 

14. For Table 4, justify whether the power is adequate for comparing zones without a type II error. It may be appropriate to perform an analysis at the regional level. Include in the table footer the specifications used to calculate the analysis, and explain the initials used (BLUP Ranking).

Our response: we computed the ICC in the model and it is statistically significant hence including zones as a random effect is meaningful. 

Discussion.

15. It is important to contextualize the findings and discuss whether the geographic delimitation of administrative zones truly reflects the characteristics of partner relationships. Do the 80 ethnic groups in Ethiopia align with this delimitation?

Our response: this is a very important point of view. We used the DHS dataset and the shapefile of the zones, we tool the covariates from the DHS dataset and the random component (zone) from the shapefile. Neither the shapefile nor the DHS dataset consist the 80 ethnic group information. However, we improved the discussion section now as per your comment. 

16. Does partner violence differ within these administrative zones when considering the social determinants model?1 It is possible that some subsets of these zones share common characteristics (social norms, cultural aspects, population structure, religion, socioeconomic level, etc.) and should be analyzed together and contextualized. Additionally, there may be differences within some of these zones.

Our response: thank you so much, this is another insight. The multilevel spatial models can be used to see the within and between variabilities. But in this paper, we focused on the multilevel aspect and we didn’t see the spatial modeling. 

1Heise L, Greene ME, Opper N, Stavropoulou M, Harper C, Nascimento M, et al. Gender inequality and restrictive gender norms: framing the challenges to health. Lancet. 2019;393(10189):2440-54.

17. It is important to include the study limitations. Some potential limitations could be that the study is based on a cross-sectional design, which only allows for the identification of associations. Additionally, the violence data is self-reported, which may introduce reporting bias due to desirability bias, with individuals more likely to report severe or recent violence,

What happen if there was another person when they asked about intimate partner violence? If there was no privacy, it was recorded in the survey; if so, were they excluded from the analysis?

Our response: thank you done 

18. At Ethiopy DHS the question about violence it is reported who perpetrated the violence. Is it possible that the violence was committed by a former partner and not by his current partner? If there is that possibility, include it in limitations.

Our response: this is a very critical issue. But we considered lifetime violence and a woman may have one or more violence by their current or previous partner. This is because the questionnaire in the model asks about lifetime violence. 

19. Are there any other self-reported variables that may be influenced by desirability bias?

Our response:  

In response to comments from Reviewer #2:

1. Reviewer #2: Please check the formatting of the manuscript including spacing, punctuation, and grammar.

Our response: thank you, we corrected it as per your suggestions. 

Methods:

2. The section should clearly mention that the analysis adjusts for the complex sampling design of the EDHS data by applying sampling weights. It is important to reiterate that all estimates presented are weighted to account for the sampling design

Our response: thank correction has been made in the revised form of the manuscript. 

3. The selection and justification of covariates included in the model are not adequately detailed.

Our response: we added the conceptual framework and justified it from different literature. 

Results:

4. The interpretation of the spatial distribution results could be more detailed linking them to underlying factors (for example, cultural, socio-economic, and regional factors).

Our response: we treated the covariates as fixed effects and the zones as random components. We see the prevalence distribution of different forms of IPV across the zones, hence our objective was not to see the spatial effects of covariates on IPV across zones. This is a good research gap for further analysis. 

Discussion:

5. While the discussion references existing literature, it could benefit from a more systematic comparison of the study’s findings with those of previous studies in detail. Currently the discussion is brief, but it could be strengthened by a more in-depth analysis of the findings, a clear connection to the Ethiopian context, and should delve into the reasons behind these findings comparing with the previous literature. Furthermore, the strengths and limitations of the study are not clearly outlined. 

Our response: done

---

## [Editor Report · Decision Letter 1]

23 Aug 2024

Spatial Distributions and Determinants of Intimate Partner Violence among Married Women in Ethiopia across Administrative Zones

PONE-D-23-29229R1

Dear Auhtor,

We’re pleased to inform you that your manuscript has been judged scientifically suitable for publication and will be formally accepted for publication once it meets all outstanding technical requirements.

Kind regards,

Meesha Iqbal

Academic Editor

PLOS ONE

---

## [Editor Report · Acceptance letter]

4 Nov 2024

PONE-D-23-29229R1 

PLOS ONE

Dear Dr. Fenta, 

I'm pleased to inform you that your manuscript has been deemed suitable for publication in PLOS ONE. Congratulations! Your manuscript is now being handed over to our production team.

Kind regards, 

on behalf of

Dr. Meesha Iqbal 

Academic Editor

PLOS ONE